# Does Remote Work Make People Happy? Effects of Flexibilization of Work Location and Working Hours on Happiness at Work and Affective Commitment in the German Banking Sector

**DOI:** 10.3390/ijerph19159117

**Published:** 2022-07-26

**Authors:** Timo Kortsch, Ricarda Rehwaldt, Manon E. Schwake, Chantal Licari

**Affiliations:** 1Department of Health and Social Work, IU International University, 99084 Erfurt, Germany; 2Department of Business and Management, IU International University, 99084 Erfurt, Germany; ricarda.rehwaldt@iu.org; 3Department of Education & Social Sciences, University of Hildesheim, 31141 Hildesheim, Germany; schwake@mzhost.de; 4FELICICON GmbH, 13127 Berlin, Germany; chantal.licari@hotmail.de

**Keywords:** remote work, happiness, commitment, banking sector, quasi-experimental design, work design, new work

## Abstract

(1) Background: In view of the advancing digitalization of the German banking sector, offering remote work can be an opportunity for banks to meet changing customer and employee needs at the same time. It allows flexible consultations at changing locations and, due to the high degree of autonomy, it also increases motivation, meaningfulness, happiness at work, and commitment. (2) Methods: This study used a quasi-experimental design to investigate how remote work affects happiness at work and affective commitment among employees in a German public bank. Therefore, two groups of customer advisors were examined, who work either remotely (N = 32) or stationary (N = 110) at similar tasks. (3) Results: The group comparisons show significantly higher values overall on three of the investigated four happiness dimensions (“meaningfulness”, “self-actualization”, and “community professional”) for employees in the remote group. Commitment also differs, as employees in the remote group show significantly stronger commitment. The quantitative results were confirmed by qualitative interviews. (4) Conclusions: By investigating the positive effects of remote working, this study shows new findings on what is likely to be a growing design form of New Work in the future. The study provides evidence that self-selected work environments and working hours offer an opportunity to make work more conducive to happiness—even in a sector that still undergoes significant shifts.

## 1. Introduction and Theoretical Background

Digitalization is fundamentally changing the world of work. One industry that is intensively addressing digitalization strategies is the banking sector [1]. The acceleration of technological change and fierce competition have put companies under pressure in recent years [2,3,4]. Previous studies on digitalization in the banking sector have focused either on the strategic level or on the customer perspective [5]. In addition to these two perspectives, banks should also integrate the employee perspective, especially with regard to happiness at work and commitment. Digitalization often leads to a disproportionate increase in work intensity in the banking sector, which has been in upheaval since the turn of the millennium due to several crises (including the 2008 financial crisis) [6]. At the same time, the traditional local branch network is being significantly thinned out in the German banking sector in the course of massive structural change, which is fundamentally changing working conditions for employees [7]. In addition, optimization and cost-cutting developments are leading to a decline in the attractiveness of the banking sector as an employer [8], while employees desire working conditions that create meaning and allow for self-fulfillment [9] and enable happiness at work overall [10,11]. In this respect, banks are faced with the dual challenge of responding to changing customer needs on the one hand and being interesting for employees in order to retain them on the other. Making work more flexible can therefore be an opportunity for the banking sector to meet both customer and employee needs: Consultations can take place at individual locations (e.g., at the customer’s home) and at flexible times instead of being bound to fixed opening hours and locations (branches). This is also referred to as remote (e-)work, which means “work being completed anywhere and at any time regardless of location and to the widening use of technology to aid flexible working practices” [12] (p. 529). Remote work is associated with different dimensions of well-being: the associations of remote work to the affective and professional dimension of well-being seem to be, in general, positive [13]. The related concept of telecommuting (which is primarily aimed at the free choice of work location) is also positively linked to job satisfaction [14]. 

The spatial effect of the self-chosen place of work (as opposed to the predetermined office workplace) takes on an important significance: Features such as stimulation, affordances, and recreational qualities can have a positive effect on the psyche [15] and changing environments can have a restorative effect on attention [16,17]. Despite the increasing prevalence of remote work, its long-term consequences, e.g., on work emotions and emotional commitment of employees, have not yet been sufficiently investigated. The aim of the present study is therefore to examine the specific effects of remote work on bank employees’ feelings of happiness at work and affective commitment. For this purpose, a qualitative preliminary study and a quantitative study with a quasi-experimental design were used, in which two groups of employees (branch work vs. remote work) of a German public bank were examined. The results are intended to provide indications of the extent to which remote work is a possible flexible form of work design that not only takes into account customer interests but also meets the needs of employees for self-fulfilling work and can thus be recommended for the banking sector and other sectors that face structural challenges.

### 1.1. Challenges of the German Banking Sector

The global economy was hit hard by the COVID-19 pandemic in 2020. GDP growth in Germany fell by 4.8% in 2020, ending a ten-year growth phase and causing the deepest recession since the financial crisis in 2008 [7]. German industry was hit hard by the downturn in the global economy, as it is heavily involved in global trade and production chains [7]. The banking sector in Germany also shows some peculiarities: First, Germany is the country with the most branches of credit institutions outside the EU, practically twice as many as the second-ranked country, Italy [7]. Second, the banking sector is divided into three pillars: the private commercial banks, the public-law banks, and the cooperative banks—which differ in terms of legal form and ownership structure. This study examined the labor situation of the second pillar: the public banking sector, which accounts for 26% of banking system’s total assets (cf. [7,18]). This is organized as a corporation under public law and restricts its activities and decisions (e.g., the regional principle, according to which these banks may be only in its business territory) to a delimited area and is controlled by municipal owners. Apart from their regional focus, their actual business does not differ from that of private commercial banks [7]. Differences are found, however, in consolidation behavior. In all three pillars, the number of banks has declined sharply, by 59% since 1995. This consolidation has largely taken place within the existing pillars. However, in the savings bank and cooperative sector (as opposed to mergers in the private sector), consolidation has often been the result of stress rather than proactive business considerations. These changes are also being felt by employees in changing working conditions. 

The transformation of the German banking sector [19] is driven, in particular, by advancing digitalization. Digitization means that financial services are becoming more interchangeable and are therefore subject to strong competitive pressure, especially from purely online banks [19]. Bank employees are particularly burdened by this, as job insecurity, work intensity, and flexibilization are increasing [19], leading to additional psychological stress [20]. As a result, the banking sector continues to lose its attractiveness as an employer [8]. 

Opportunities for customers and advisors to meet face-to-face have become rarer, as the number of bank branches in Germany was already reduced by more than a third between 2001 and 2013 [19]. In 2020, Germany again saw one of the largest declines in the number of branches in the EU [7]. 

This increases the pressure on bank employees to be present and effective in those rare moments of customer contact. Therefore, new ways of direct customer contact are being sought. One of these ways—remote work—was examined in this study. There is an urgent need for today’s banking market to create new ways of interaction and, thus, drive workplace design [19] to remain attractive as an employer.

### 1.2. Remote Work as an Alternative to Work Design

Remote work can be seen as an alternative form of work design where employees are performing “tasks away from their primary offices, using information and communication technologies (ICTs) to interact with others inside and outside of their organization” [20] (p. 165). Mainly due to the corona pandemic [20,21,22], remote work has become a regular part of work for more than one-third of all employees in Germany [23]. Even though the process of remote work was accelerated by the COVID-19 pandemic, the proportion of remote workers already tripled from the 1980s to the 2010s [24,25]. This way, commutes can be reduced, and the carbon footprint of employees decreases [25]. In contrast to solely working at home, remote workers can work in changing environments (e.g., on the balcony, in a café, in the park, at the customer’s premises, etc.), each of which can develop its own spatial effects. One possible explanation for the positive effect of changing environments is offered by the attention-recovery theory [16,17]: according to this, the change of location made possible by remote work leads to more varied environments, which have a recovery effect. Empirical studies provide evidence of the importance of the environment for well-being. People in natural environments are happier [26,27]. The very pathways associated with changes of place can have a positive effect: changes of location increase the likelihood of moving around in natural environments, and this, in turn, has been shown to promote well-being [28].

Remote work can also increase motivation [29] and sense of meaning [30] due to its high autonomy [14], and it can also increase happiness at work [11] and retention [31]. Studies have shown that the perception of a self-determined environment was associated with higher autonomy and, thus, higher motivation [32,33]. If the choice of work environment in the context of remote work is made strategically to align with personal needs and goals (e.g., working in a café because the feeling of having people around is liked), this can be understood as structural job crafting [34]. Job crafting reduces work stress and can thus further enhance the positive experience of work [35]. 

### 1.3. Effects of Remote Work on Happiness at Work 

Remote work is associated with several positive effects on employees. For example, remote work offers the opportunity to break routines or escape managerial control [36]. It has a positive impact on the work-life balance [37] and can lead to better performance and increased job satisfaction [14,22,37,38]. It thus provides space for the development of positive emotions and happiness at work. 

Studies show a variety of beneficial effects of positive emotions at work: people in a positive mood show higher rates of cognitive performance [39], can combine work steps more efficiently [40], and can make decisions faster [41]. They are more creative and exhibit high problem-solving skills [42]. Studies also show that happy employees have significantly fewer sick days [43,44]. The corporate relevance of happiness at work is justified, among other things, by the fact that happiness leads to increased customer loyalty and productivity [45]. Both aspects are of great importance, especially in the banking sector, as corporate developments focus on simplifying and standardizing processes. 

According to Rehwaldt (2017) [11], happiness at work is determined by the factor’s meaningfulness, self-actualization, and community. Meaningfulness arises when employees feel they are contributing to the bigger picture or helping someone [11]. Remote work could strengthen employees’ sense of meaning through on-site appointments with customers, as the feeling of supporting and helping comes to the fore. Self-actualization is promoted through the scope for action and the use of one’s own skills and potential [11]. The factor of self-actualization also moves further into focus in the context of New Work [9] and New Learning [46] in the sense of doing “what you really really want”. Freedom of action concerning work location and working hours supports self-actualization in the context of work. The community factor arises, among other things, through the social interaction of team members on a professional (rather task-related) and a familiar (rather emotional) level [11]. Task-related interactions are also possible digitally (albeit with limitations), but the emergence of trust and team cohesion is primarily possible when real and regular encounters also take place [47]. For example, teams who work together virtually perceive a lack of important information, namely team feedback [48,49]. One recent study on the effects of remote work showed that the communication between remote workers becomes more asynchronous, the communication media are less rich (e.g., email) in terms of media richness, and collaboration becomes more siloed [50].

Building on these considerations, we therefore assume the following:

**Hypotheses** **1a** **and** **1b.**
*Remote-working employees have higher values on the happiness subscales self-actualization (1a) and meaningfulness (1b) than branch sales employees.*


**Hypothesis** **1c** **and** **1d.**
*Remote-working employees have lower scores on the happiness subscales community professional (1c) and community familiar (1d) than branch sales employees.*


### 1.4. Effects of Remote Work on Commitment 

To be successful, companies are not only dependent on motivated and committed employees, but also on keeping the employees and their know-how within the company. Therefore, the importance of affective commitment is becoming increasingly important for companies. There are at least two reasons for this: On the one hand, the dependence of employees on their companies is decreasing, and, thus, measures to increase the retention of qualified personnel are coming into focus [51]. Secondly, affective commitment promotes performance and creativity [52] and the willingness to work for the company [53]. For employees with high affective commitment, their work also represents a part of their own identity [54].

Meta-analytical evidence suggests that telework, a related construct to remote work, is positively associated with commitment [55,56]. One reason might be the flexibility of remote work [38]. Flexibilization measures reduce absenteeism, i.e., absence due to illness [57], and are also of great importance to job applicants [58]. Meta-analytical studies have shown that flexibility in one’s work schedule reduces the intention to change jobs and, thus, increases employees’ loyalty to the company [31]. Bjärntoft et al. (2020) [59] cite an increase in perceived autonomy and flexibility—which are also fundamental in intrinsic motivation processes [29]—as the reason for the increased work-life balance through job flexibility. Affective commitment to the company could be increased through freedom in work design, e.g., through remote work [60]. We, therefore, assume the following:

**Hypotheses** **2a–2c.**
*Remote-working employees show higher affective commitment to the company (2a), to the job (2b), and the team (2c) than branch sales employees.*


## 2. Materials and Methods

### 2.1. Pre-Study

To verify the hypotheses for the company under investigation, three qualitative interviews were conducted for each sub-sample. This not only enabled a practical confirmation of the hypotheses derived theoretically so far, but also provided insights into possible causes and causalities. The total of six interviews lasted each between 20 and 35 min. One manager and two employees from each subgroup (i.e., branch and remote) were interviewed. The employees were one experienced person and one newcomer (joining the company about a year ago) to the department. The semi-structured interviews were based on a guideline that focused on the relationship with the manager and the colleagues, as well as the feeling of happiness and the challenges at work. The interviews served to gain a deeper insight into the differences between employees working in the branch and those working on the move.

The impressions from the interviews can be summarized in categories as follows: Employees in the remote group primarily strived for personal autonomy and room for maneuver and emphasized aspects of self-actualization, such as the realization of ideas and the opportunities to show initiative and to contribute ideas. Similarly, remote workers emphasized the community aspects such as support and exchange among colleagues and the improved opportunities to build an individual and authentic customer relationship. This can be seen, for example, in the quotes in Table 1.

The members of the branch group primarily emphasized the communal relationships that create a sense of belonging through fixed communication structures and fixed affiliations and locations. This is reflected in the sample quotations in Table 2.

Overall, the assumption was confirmed that employees in the remote group would report differently about their work, have a different mindset, and appear more committed overall.

### 2.2. Study Design and Sample

Employees from a public bank who either had fixed working hours and a fixed workplace due to branch opening time (branch group) or worked flexibly in terms of location and time (remote group) were surveyed through a questionnaire. This created two natural, non-manipulated groups that were to be compared with each other, as is typical for a quasi-experimental design. The employees in both groups performed the same tasks in terms of content, but they differed greatly in terms of their working conditions. 

In the remote group, the survey was conducted via e-mail, with the completed questionnaires handed in by e-mail or as a printout directly in the office. Initially, there was a response of N = 17. To reach the relatively small sub-sample of remote workers as completely as possible, the opportunity to take part in the survey was given again at an annual kick-off event. Here, another N = 15 employees took part. Thus, a total of N = 32 employees from the remote group took part (response rate of 86.5% of all employees who work remotely). In the branch group, the survey was conducted exclusively by e-mail due to the wide distribution of branches. In the branch group, employees could return the completed questionnaire anonymously by internal post or by e-mail. A sample of N = 110 employees was reached (response rate: 23.2%). Both groups were similar in terms of sample characteristics (with small differences; see Table 3). Among the employees in the branch group, there were slightly more women (54.55% vs. 46.88%); they were, on average, slightly older (the largest age groups: 49.09% in the 46–55 age group vs. 43.75% in the 36–45 age group), and more than two-thirds (71.82% vs. 56.25%) of the employees had more than 20 years of service; the employees in the remote group worked more often in full-time.

### 2.3. Instruments

*Happiness at work.* Happiness at work was measured with the validated HappinessandWork-Scale [10]. The HappinessandWork-Scale measures the four formative happiness factors with three items each: *meaningfulness* (e.g., “With my work I actively contribute to the well-being of others”), *self-actualization* (e.g., “In my work I have a lot of freedom”), *community professional* (e.g., “Even in tense situations no one in our company shifts responsibility to someone else”), and *community familiar* (e.g., “If I have private problems, I discuss them with my colleagues”). These are surveyed by using a five-point Likert scale, from 1 = “do not agree at all” to 5 = “agree completely”. Reliabilities ranged from α = 0.66 (subscale *community familiar*) to α = 0.83 (subscale *meaningfulness*).

*Commitment.* Affective commitment was measured by using the COMMIT questionnaire [61]. Each of the selected foci is measured by using three items: *Company* (e.g., “I feel a strong sense of belonging to the [company name]”), *Team* (e.g., “I feel close to my team”), and *Occupation* (e.g., “I enjoy my current job”). These are recorded by using a five-point Likert scale, from 1 = “strongly disagree” to 5 = “strongly agree”. The reliabilities were between α = 0.86 (subscale *commitment to the company*) and α = 0.91 (subscale *commitment to the team*).

## 3. Results

### 3.1. Statistical Analyses

The program JASP (version 0.14.1) [62] was used for hypothesis testing and the associated calculations. In view of the different group sizes, both groups were compared by using non-parametric tests (Mann–Whitney U test).

### 3.2. Results for the Hypotheses

Due to the significantly different group sizes, Mann–Whitney U tests were conducted to compare the two groups. The tests revealed significant differences between the two groups for the happiness scales self-actualization (W = 2790.00, *p* < 0.001), meaning (W = 2695.00, *p* < 0.001), and community professional (W = 2655.50, *p* < 0.001); in each case, the values in the remote group were significantly higher (see Table 4). The effects can be classified as large [63]. On the other hand, there was no significant difference in the community familiar scale (W = 2120.00, *p* > 0.05). Thus, hypotheses 1a and 1b could be confirmed, while hypotheses 1c and 1d could not be confirmed.

The tests showed significant differences between the two groups for the three foci commitment to the company (W = 2308.00, *p* < 0.01), commitment to the job (W = 2597.50, *p* < 0.001), and commitment to the team (W = 2356.00, *p* < 0.01); the values in the remote group were significantly higher (see Table 5). The effects can be classified as medium to large [63]. Hypotheses 2a, 2b, and 2c are thus confirmed.

## 4. Discussion

Digitalization is leading to a steadily increasing flexibilization of work and once again raises the question: Where and how do people want to work? For the German banking sector, this study shows that remote work can be a way to offer employees working conditions that promote happiness and commitment, while at the same time responding to changing customer needs. Self-determination of work location and working hours plays a central role in this. The study was a quasi-experimental design that showed, in line with the hypothesis, that employees in the remote group had significantly higher values for two of the four happiness-promoting factors (“meaningfulness” and “self-actualization”) compared to the branch group. The interviews from the preliminary study concretized what is meant by this, namely that personal autonomy and scope for action are seen as very central. Thus, remote work as a work design measure seems to have further triggered the individual “job crafting” [34] of the employees, which can then be perceived as a resource, especially in the context of additional challenges, e.g., due to the corona pandemic [22].

Surprisingly, there were also higher values for the community-related happiness factors in the group of remote workers—once significantly (“community professional”) and once tendentially (“community familiar”); this fundamentally contradicted the hypotheses that assumed a significantly lower value in the remote group. The interviews also contradicted the hypotheses regarding the community factors; a community was also experienced in the remote group. One reason for this may be the ingroup–outgroup effect (cf. Reference [64]): The remote workers perceive themselves as a group due to the special working conditions, and, therefore, they also support each other. This may have been additionally reinforced by a self-selection effect in the remote group: it is possible that those who chose to work remotely are similar to each other (e.g., sample characteristics indicate a lower age than in the branch group), and this may have further reinforced the group feeling. Thus, a sense of community emerged among the remote workers to the same or a higher degree than among the employees in the branch group. The commitment values were significantly higher for the remote group than for the branch office group for all three foci: company, job, and team. At the same time, however, it was found that the levels of the happiness factors and the commitment facets were also in the positive range of the scale in the branch office group and that the commitment to the company was even very pronounced. In this respect, remote work seems to be a way of generating positive effects, even among already happy and committed employees.

### 4.1. Theoretical Implications

Several theoretical implications follow from this study. First, a qualitative preliminary study and a quasi-experimental design demonstrated positive effects of remote work on happiness and commitment. Thus, this study complements previous findings on the effects of remote work (see References [12,13], for example), with findings on two relevant constructs. At the same time, the study expands the findings with the German banking sector, a sector that has already been confronted with massive upheaval since the turn of the millennium. However, most research focused on the financial impacts of these changes (see Reference [18], for example). Thus, the banking sector can also be seen as exemplary for sectors that are undergoing major upheavals (e.g., the automotive sector) or where upheavals are imminent. 

Secondly, the study supports the theory of happiness at work [11] with further empirical findings. In particular, the different results on the community factors community professional, and community familiar seem to be fruitful for further research. According to this, it is not only important for the experience of happiness in a professional context whether one experiences social inclusion [29], but also what quality this social inclusion has—according to Rehwaldt (2017) [11], both the opportunity for professional exchange and the experience of personal closeness are optimal. 

Thirdly, this study thus shows that remote work as a work design measure combines employee and company interests and appears to be a profitable measure, at least regarding the dimensions of happiness and commitment that were examined. In terms of social exchange [60], remote workers experience more happiness at work, and, at the same time, their commitment to the company increases.

### 4.2. Practical Implications

For companies, the study provides insightful findings regarding the use of remote work, as remote work has positive effects on desired outcomes compared to traditional non-remote work. 

Firstly, remote work seems to be an effective tool for promoting the happiness factors of meaningfulness, self-actualization, and community professional (cf. Reference [11]), as the remote group had significantly higher scores here. The effects of the three happiness scales were large. Studies show that varied work environments have a positive effect on well-being [16,17] and natural environments contribute to an increase in happiness [26,27] In this respect, employers can enhance the happiness perception of employees by using remote work as a health-promoting job design measure in a targeted manner and support it, for example, with the training of job crafting strategies [34]. However, personal interactions should be offered, especially for the aspect of trust (familiarity factor), but also to avoid feelings of loneliness [38]. One possibility for this would be weekly meetings, but also methods such as “Working Out Loud” [65], in which groups meet weekly over a longer period of time to work on a self-selected development topic.

Secondly, remote work appears to be an effective tool for increasing affective commitment. The study showed that remote work promotes commitment to the company, to the job, and to the team. Remote work can therefore be an interesting way for companies to remain an attractive employer and to gain a boost in attractiveness [66]. Especially when space design in offices is not possible, remote work offers a variant of designing a healthy working environment. This can additionally be combined with the establishment of digital exchange rooms to reduce the negative effects on the community dimension (professional and familiar) [25]. Depending on the business sector, it is possible, for example, to offer “daily scrums” [67] or digital coworking. In addition, community-building measures such as team-building events can be offered to promote getting to know each other and to allow familiarity to develop. The feeling of happiness can also be further enhanced through relationship-based leadership [59].

### 4.3. Limitations and Further Research

Even though the study provides new insights into the effect of remote work in the banking sector, there are some limitations. While positive effects of remote work could be demonstrated, they are associated with challenges that should be further investigated in additional studies to gain further clarity on the effects of remote work. It is known that remote work favors a blurring of the boundary between work and private life (see References [22,25,68]). This is associated with permanent accessibility [69] and working after hours [70], which will continue to increase due to the increasing virtuality of collaboration (see [47,71]). Higher virtuality may also decrease creativity [72] and increase silo thinking [50]. Furthermore, certain leadership behaviors (e.g., intrusive leadership) heavily interfere with remote work [70]. However, it seems that it is not the extent of a flexible work arrangement such as remote work itself but the associated job characteristics (e.g., autonomy) that can cause negative effects [73]. It is, therefore, all the more important to design remote work well to minimize risks. 

A methodological limitation of the study is that the data collection of the remote group had to take place partly within the framework of a team event, as a result of response problems. Although this significantly increased the response rate, this team event could have had additional positive influences on the happiness experience of the remote group. Even if the character of the event was rather formal, the employees of this group met each other in this setting and were, thus, possibly able to experience a sense of community—because the promotion of interpersonal relationships is one of the four confirmed effective components of such team events [74]. However, the fact that no significant differences were found in the dimension “community familiar” speaks against this confounding effect. On this dimension, with the highest affectivity of the four dimensions, effects would have been most likely to be expected through such an intervention [74].

Other things to note are the different group sizes and the slightly different sample characteristics. For example, the remote workers were all employed full-time and tended to be younger than the employees in the branch. Here, one cannot exclude a certain self-selection of working conditions (remote vs. branch), which could have influenced the results. This is reinforced since workers in the remote group could choose whether to work in the branch or remotely. Even if it makes sense to let employees have a voice regarding the working conditions under which they work (in this case, remote vs. branch), this is not always possible (e.g., in the wake of the corona pandemic or due to cost-cutting measures). Therefore, the present study should be replicated with a different sample and different methods (e.g., randomized group assignment or control for baseline levels of happiness before the start of the intervention) in the future. However, current data sets from the corona period, in contrast to this study, may be influenced by confounding factors that occurred as a result of the COVID-19 pandemic, such as increased anxiety about one’s health.

## 5. Conclusions

In view of the advancing digitalization, work will continue to become more flexible in the future in order to better take into account employee and customer interests. This means that the special flexible form of remote work will also increase. Through the present results, it could be shown that the increasing flexibilization and growing decision-making possibilities have a positive influence on the perceived happiness at work. The commitment of remote workers (toward the team, the company, and the job) is also growing. Particularly due to the corona pandemic, it is of the highest social relevance to further research the influences of remote working, which also includes the home office.

## Figures and Tables

**Table 1 ijerph-19-09117-t001:** Remote group (shortened version).

Category	Sample Quotes
Personal autonomy and scope for action	“They let me work here in a self-determined way” (Employee). As long as I make my numbers, I could also start work at 4pm and be at home by then, baking cakes and doing the laundry. That’s up to me” (Employee). “[…] Organize themselves at home, go to the customer” (Manager).
Innovation, initiative, ideas	“The management is also always ready to say, we’ll try something new. And if it works, then it’s great. If it doesn’t work, you just have to do it differently” (Employee). “So what I can say that makes remote counselling work is the issue of personal responsibility and initiative” (Manager).
Customer orientation or relationship orientation	“Super advantageous for the client, because you come to his home. So logically, different atmosphere as well. The customer feels more comfortable” (Employee). “Because when you leave the branch, everything is quite regimented and uniformed, that’s my experience. And being different is stupid. Here, being different is intentional. Because we have to adjust to customers, and they don’t want soldiers, they want authentic godfathers at eye level” (Manager).
Support and exchange with colleagues and manager (community)	“Despite all that, we exchange a lot of information and say, ‘Hey, how would you approach this case? Or can I give you some feedback?’ We do that a lot, especially the management” (Employee). “In addition, I also have a face-to-face situation with each individual employee every week” (Manager).

**Table 2 ijerph-19-09117-t002:** Branch group (shortened version).

Category	Sample Quotes
Belonging	“So I need that home feeling here” (Employee). “But today I am also satisfied that I am here. Why? Because I am a permanent member of the team here and have a fixed location” (Employee).
Fixed communication structures	“I see my manager every day. We talk every day. […] But we also help each other” (Employee). “I have a meeting with all my staff every week that is fixed and scheduled” (Manager).

**Table 3 ijerph-19-09117-t003:** Distribution of sample characteristics in the remote and branch groups in comparison.

Characteristic	Branch Group (N = 110)	Remote Group (N = 32)
Gender (% female)	54.55%	46.88%
Proportion of managers	17.27%	18.75%
Age groups:		
18–25 years	5.45%	3.13%
26–35 years	13.64%	28.13%
36–45 years	20%	43.75%
46–55 years	49.09%	21.88%
56–65 years	11.82%	3.13%
Employment relationship:		
Full-time, permanent	75.45%	96.88%
Full-time, limited	1.82%	3.13%
Part-time, unlimited	18.18%	0.00%
Part-time, limited	4.55%	0.00%
Length of service:		
0–5 years	2.73%	3.13%
5–10 years	7.27%	18.75%
10–15 years	10.91%	12.5%
15–20 years	7.27%	9.38%
Over 20 years	71.82%	56.25%

**Table 4 ijerph-19-09117-t004:** Descriptive statistics and group comparisons on happiness factors.

Happiness Factor	Group	*N*	*M*	*SD*		Cohen’s *d*	Group Comparison	
							*W*	*p*
Self-actualization	Remote	32	4.13	0.51		1.03	2790.00	<0.001
	Branch	111	3.37	0.78				
Meaningfulness	Remote	32	4.29	0.59		0.95	2695.00	<0.001
	Branch	111	3.61	0.75				
Community professional	Remote	32	4.26	0.40		0.88	2655.50	<0.001
	Branch	111	3.64		0.77			
Community familiar	Remote	32	3.98	0.64		0.36	2120.00	>0.05
	Branch	111	3.70		0.82			

Notes: Group comparisons due to large differences in group size using the nonparametric Mann–Whitney U test.

**Table 5 ijerph-19-09117-t005:** Descriptive statistics and group comparisons on the commitment scales.

Commitment Focus	Group	*N*	*M*	*SD*	Cohen’s *d*	Group Comparison	
						*W*	*p*
Commitment to the company	Remote	32	4.49	0.92	0.41	2308.00	<0.01
	Branch	111	4.12	0.90			
Commitment to the job	Remote	32	4.16	1.21	0.54	2597.50	<0.001
	Branch	111	3.66	0.81			
Commitment to the team	Remote	32	4.41	0.48	0.65	2356.00	<0.01
	Branch	111	3.80	1.02			

Notes: Group comparisons due to large differences in group size, using the nonparametric Mann–Whitney U test.

## Data Availability

The data presented in this study are available upon request from the corresponding author. The data are not publicly available in order to prevent drawing conclusions about participating subjects and the participating bank.

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
