# Peer review of "Does Remote Work Make People Happy? Effects of Flexibilization of Work Location and Working Hours on Happiness at Work and Affective Commitment in the German Banking Sector"

_ijerph, 2022, doi:10.3390/ijerph19159117_

Round 1

Reviewer 1 Report

Title: Does remote work make people happy? Effects of Flexibilization of Work Location and Working Hours on Happiness at Work and Affective Commitment in the German Banking Sector

This study investigated the effect of remote work on happiness at work and the commitment of the employees of the German Banking Sector. I think this study is very timely for understanding the effects of remote work, especially given the expanding use of work design due to the COVID-19 pandemic and digitization. Although this study has several advantages, it also has some issues that need to be addressed to make the manuscript better. Here are some issues to address. 

First, hypotheses 1a and 1b are about the positive effects of remote work, and hypotheses 1c and 1d are about the negative effects of remote work. While the rationale for hypotheses 1a and 1b was given relatively sufficiently, the rationale for hypotheses 1c and 1d was not sufficient. In the paper, only one sentence is presented as the basis for hypotheses 1c and 1d: “Task-related interactions are also possible digitally (albeit with limitations), but the emergence of trust and team cohesion is primarily possible when real and regular encounters also take place.” Therefore, the author(s) should elaborate more to provide more detailed explanations to provide sufficient justification for hypotheses 1c and 1d. 

Second, on page 4, the following sentences are not consistent. “In order to be successful, companies are not only dependent on motivated and committed employees, but also on keeping the know-how within the company. Therefore, the importance of affective commitment is becoming increasingly important for companies.” -> the first sentence emphasizes “keeping the know-how within the company” and the second sentence emphasizes affective commitment. Given the overall purpose of this study, it is better to rephrase the first sentence.  

Third, the subject of this study is public banks in Germany, which account for 26% of the total assets of the banking system. However, private commercial banks represent the largest segment, which accounts for 40% of the banking system’s total assets. Is there any reason why the second-largest segment was selected as the subject of this study? If there are specific reasons, it is better to explain them. 

Fourth, the results of the interviews were summarized on page 5, and the results seem to be different from the initial assumption that remote work has a negative effect on the community factor. The results of the survey were also consistent with the results of the interviews, because, irrespective of statistical significance, the community factor was higher in the remote workgroup. There was no significant difference in community familiarity between the remote workgroup and branch group, but the average of the community familiar for the remote group was higher than the branch group. Therefore, the results for the community factor are contrary to the initial arguments. However, in the discussion section, there is not enough discussion on the reasons for these results Because the results differed from the original arguments presented, the authors should elaborate more on why these results occurred.

Finally, it seems necessary to discuss whether there are criteria for allowing remote work for selecting remote workers because remote working employees appear to represent a fraction of the total workforce. This is because those selected as remote workers might feel a high sense of happiness or commit themselves regardless of working conditions. 

Author Response

Dear Reviewer,

Thank you very much for your valuable comments on our manuscript, to which we are very happy to comment below.

Comments and Suggestions for Authors

Title: Does remote work make people happy? Effects of Flexibilization of Work Location and Working Hours on Happiness at Work and Affective Commitment in the German Banking Sector

This study investigated the effect of remote work on happiness at work and the commitment of the employees of the German Banking Sector. I think this study is very timely for understanding the effects of remote work, especially given the expanding use of work design due to the COVID-19 pandemic and digitization. Although this study has several advantages, it also has some issues that need to be addressed to make the manuscript better. Here are some issues to address.

First, hypotheses 1a and 1b are about the positive effects of remote work, and hypotheses 1c and 1d are about the negative effects of remote work. While the rationale for hypotheses 1a and 1b was given relatively sufficiently, the rationale for hypotheses 1c and 1d was not sufficient. In the paper, only one sentence is presented as the basis for hypotheses 1c and 1d: “Task-related interactions are also possible digitally (albeit with limitations), but the emergence of trust and team cohesion is primarily possible when real and regular encounters also take place.” Therefore, the author(s) should elaborate more to provide more detailed explanations to provide sufficient justification for hypotheses 1c and 1d.

#Response to the comment:

Thank you very much for your valuable comment. We provided additional arguments to justify hypotheses 1c and 1d. We have added the following sentences in this regard on page 4:

“For example, teams who work together virtually perceive a lack of important information, namely team feedback [48,49]. One recent study on the effects of remote work showed that the communication between remote workers becomes more asynchronous, the communication media are less rich (e.g., email) in terms of media richness and collaboration becomes more siloed [50].

Second, on page 4, the following sentences are not consistent. “In order to be successful, companies are not only dependent on motivated and committed employees, but also on keeping the know-how within the company. Therefore, the importance of affective commitment is becoming increasingly important for companies.” -> the first sentence emphasizes “keeping the know-how within the company” and the second sentence emphasizes affective commitment. Given the overall purpose of this study, it is better to rephrase the first sentence. 

#Response to the comment:

Thank you for the comment. We rephrased the sentence to put more emphasizes on the employees: “In order to be successful, companies are not only dependent on motivated and committed employees, but also on keeping the employees and their know-how within the company. Therefore, the importance of affective commitment is becoming increasingly important for companies.”

Third, the subject of this study is public banks in Germany, which account for 26% of the total assets of the banking system. However, private commercial banks represent the largest segment, which accounts for 40% of the banking system’s total assets. Is there any reason why the second-largest segment was selected as the subject of this study? If there are specific reasons, it is better to explain them.

#Response to the comment:

Thank you very much for your valuable comment. With the wording, we have put too much focus on asset shares, although it is rather the specifics of this group that are interesting: they are subject to legal restrictions that limit their actions, so they have to look for other options. For example, the regional principle applies to savings banks, according to which a savings bank may only operate in its business territory. We have therefore deleted the sentence in question and made additions to the wording to better explain the selection. See page 2:

“This study examines the labor situation of the second pillar: the public banking sector, which accounts for 26% of banking system’s total assets [cf. 7,18]. This is organized as a corporation under public law and restricts its activities and decisions to a delimited area (e.g., the regional principle, according to which these banks may only in its business territory) and is controlled by municipal owners.”

Fourth, the results of the interviews were summarized on page 5, and the results seem to be different from the initial assumption that remote work has a negative effect on the community factor. The results of the survey were also consistent with the results of the interviews, because, irrespective of statistical significance, the community factor was higher in the remote workgroup. There was no significant difference in community familiarity between the remote workgroup and branch group, but the average of the community familiar for the remote group was higher than the branch group. Therefore, the results for the community factor are contrary to the initial arguments. However, in the discussion section, there is not enough discussion on the reasons for these results Because the results differed from the original arguments presented, the authors should elaborate more on why these results occurred.

#Response to the comment:

Thank you very much for your valuable comment. We have again emphasized more strongly that the results are in strong contradiction to the hypothesis. Furthermore, we have made additional theoretical considerations that could explain the result on page 9:

“Surprisingly, there were also higher values for the community-related happiness factors in the group of remote workers - once significantly ("community professional") and once tendentially ("community familiar"), which fundamentally contradicted the hypotheses that assumed a significantly lower value in the remote group. The inter-views also contradicted the hypotheses regarding the community factors; a community was also experienced in the remote group. One reason for this may be the in-group-outgroup effect [cf. 64]: The remote workers perceive themselves as a group due to the special working conditions, which therefore also supports each other. This may have been additionally reinforced by a self-selection effect in the remote group: It is possible that those who chose to work remotely are similar to each other (e.g., sample characteristics indicate a lower age than in the branch group), which may have further reinforced the group feeling.”

Finally, it seems necessary to discuss whether there are criteria for allowing remote work for selecting remote workers because remote working employees appear to represent a fraction of the total workforce. This is because those selected as remote workers might feel a high sense of happiness or commit themselves regardless of working conditions.

#Response to the comment:

Thank you very much for your valuable comment. Workers could volunteer for remote work, which may naturally lead to the selection of fellow workers who feel comfortable in this work setting. We do not know if workers engage outside of work. However, we were able to show that the HappinessandWork scale captures happiness at work and that the results can be clearly distinguished from general life happiness (Rehwaldt & Kortsch, 2022).

We added some sentences in the limitations section on page 11:

“Even if it makes sense to let employees have a voice in the working conditions under which they work (in this case: remote vs. branch), this is not always possible (e.g., in the wake of the Corona pandemic or due to cost-cutting measures). Therefore, the present study should be replicated with a different sample and different methods (e.g., randomized group assignment or control for baseline levels of happiness before the start of the intervention) in the future.”

Reviewer 2 Report

Remote working can increase job satisfaction or some type of commitment (e.g. continuance commitment), but not organizational commitment, this result is unacceptable as it contradicts our scientific knowledge in this area. According to this, the level of organizational commitment is high when there is a strong day-to-day relationship between the employees and the organization, which is a necessary (but not sufficient) condition for increasing commitment. The main problem is that the authors do not explain this contradiction convincingly and have not conducted the necessary in-depth literature search for this. In addition, in their empirical research, the number of items in the sample is so small that no generalization can be made based on the results. Finally, I think that the topic of remote work and commitment can only have such a small connection to the focus of IJERPH that I don’t think it would fit into it in terms of content.

Author Response

Dear Reviewer,

Thank you very much for your valuable comments on our manuscript, to which we are very happy to comment below. 

Comments and Suggestions for Authors:

Remote working can increase job satisfaction or some type of commitment (e.g., continuance commitment), but not organizational commitment, this result is unacceptable as it contradicts our scientific knowledge in this area. According to this, the level of organizational commitment is high when there is a strong day-to-day relationship between the employees and the organization, which is a necessary (but not sufficient) condition for increasing commitment. The main problem is that the authors do not explain this contradiction convincingly and have not conducted the necessary in-depth literature search for this.

#Response to the comment:

Thank you very much for your comment. In our hypotheses 2a-c we make assumptions about affective commitment as a certain component of organizational commitment (cf. Allen & Meyer, 1990). Regarding the claim that the statement contradicts the previous knowledge, we do not agree that the findings on organizational commitment can be boiled down to the essence presented. Rather, the findings are more diverse as relevant meta-analyses describe (e.g., Cooper-Hakim & Viswesvaran, 2005; Mathieu & Zajac, 1990; Meyer et al., 2002; Riketta, 2005). For example, according to Meyer et al. (2002), work experiences are the most important antecedent of organizational commitment in general and the component affective commitment in particular. Work experiences mean organizational support, role ambiguity, and conflict as well as justice perceptions. In terms of actual implementation in practice, a variety of practices can also be used to increase commitment. Mercurio (2015, p. 403) stated in his review that “the key high-commitment HR practices that affect levels of affective commitment can be categorized as recruitment and selection, socialization, mentoring and social networking, and training and development”. In this respect, previous findings on commitment do not contradict our assumptions and findings.  

References:

Allen, N. J., & Meyer, J. P. (1990). The measurement and antecedents of affective, continuance and normative commitment to the organization. Journal of occupational psychology, 63(1), 1-18.

Cooper-Hakim, A., & Viswesvaran, C. (2005). The construct of work commitment: Testing an
integrative framework. Psychological Bulletin, 131, 241-259.

Mathieu, J. E., & Zajac, D. M. (1990). A review and meta-analysis of the antecedents, correlates, and consequences of organizational commitment. Psychological bulletin, 108(2), 171.

Mercurio, Z. A. (2015). Affective Commitment as a Core Essence of Organizational Commitment: An Integrative Literature Review. Human Resource Development Review, 14(4), 389–414. https://doi.org/10.1177/1534484315603612

Meyer, J. P., Stanley, D. J., Herscovitch, L., & Topolnytsky, L. (2002). Affective, continuance, and normative commitment to the organization: A meta-analysis of antecedents, correlates, and consequences. Journal of vocational behavior, 61(1), 20-52.

Riketta, M. (2005). Organizational identification: A meta-analysis. Journal of Vocational
Behavior, 66, 358-384

In addition, in their empirical research, the number of items in the sample is so small that no generalization can be made based on the results.

#Response to the comment:

We are a little bit unsure what the reviewer is referring to. Regarding the number of items in the scales used, we can refer to the existing validation studies that have comprehensively examined the instruments (Franke & Felfe, 2012; Rehwaldt & Kortsch, 2022). Concerning the issue of generalization we discussed the possible problem of self-selection of the “intervention” remote work in the limitation section. However, the significant mean differences between both groups that we found can be classified as large effects according to Cohen (1992), which suggests a certain stability and generalizability of the findings.

References:

Cohen, J. (1992). A power primer. Psychological Bulletin, 112(1), 155–159. https://doi.org/10.1037/0033-2909.112.1.155

Felfe, J., & Franke, F. (2012). COMMIT: Commitment-Skalen: Fragebogen zur Erfassung von Commitment gegenüber Organisation, Beruf, Tätigkeit, Team, Führungskraft und Beschäftigungsform. Deutschsprachige Adaptation und Weiterentwicklung der Organizational commitment scale von J. P. Meyer und N. Allen [COMMIT: Commitment Scales: Questionnaire to assess commitment to organization, job, activity, team, manager, and type of employment. German adaptation and further development of the Organizational commitment scale by J. P. Meyer and N. Allen]. Huber.

Rehwaldt, R. & Kortsch, T. (2022). Was macht bei der Arbeit glücklich? Entwicklung und erste Validitätsbefunde zu einer Skala zur mehrdimensionalen Erfassung von Glück bei der Arbeit [What Makes You Happy at Work? Development and Validation of a Multidimensional Scale to Assess Happiness at Work]. Zeitschrift für Arbeits- und Organisationspsychologie A&O, 66(2), 72-86. https://doi.org/10.1026/0932-4089/a000373  

Finally, I think that the topic of remote work and commitment can only have such a small connection to the focus of IJERPH that I don’t think it would fit into it in terms of content.

#Response to the comment:

Thank you for the comment. We had sent an extended abstract to the editorial office in advance to assure us of the fit of our manuscript into the scope of the journal. The fit was confirmed by the editorial office and the guest editors of the special issue.

Round 2

Reviewer 2 Report

Thank you for your revision.